# Grounded Answers for Multi-agent Decision-making Problem through Generative World Model

**Zeyang Liu**
zeyang.liu@stu.xjtu.edu.cn

**Xinrui Yang**
xinrui.yang@stu.xjtu.edu.cn

**Shiguang Sun**
ssg2019@stu.xjtu.edu.cn

**Long Qian**
qianlongym@stu.xjtu.edu.cn

**Lipeng Wan**
wanlipeng77@xjtu.edu.cn

**Xingyu Chen**[*]
chenxingyu_1990@xjtu.edu.cn

**Xuguang Lan**[*]
xglan@mail.xjtu.edu.cn

**National Key Laboratory of Human-Machine Hybrid Augmented Intelligence**
**National Engineering Research Center for Visual Information and Application**
**Institute of Artificial Intelligence and Robotics, Xi'an Jiaotong University, Xi'an, China**

## Abstract

Recent progress in generative models has stimulated significant innovations in many fields, such as image generation and chatbots. Despite their success, these models often produce sketchy and misleading solutions for complex multi-agent decision-making problems because they miss the trial-and-error experience and reasoning as humans. To address this limitation, we explore a paradigm that integrates a language-guided simulator into the multi-agent reinforcement learning pipeline to enhance the generated answer. The simulator is a world model that separately learns dynamics and reward, where the dynamics model comprises an image tokenizer as well as a causal transformer to generate interaction transitions autoregressively, and the reward model is a bidirectional transformer learned by maximizing the likelihood of trajectories in the expert demonstrations under language guidance. Given an image of the current state and the task description, we use the world model to train the joint policy and produce the image sequence as the answer by running the converged policy on the dynamics model. The empirical results demonstrate that this framework can improve the answers for multi-agent decision-making problems by showing superior performance on the training and unseen tasks of the StarCraft Multi-Agent Challenge benchmark. In particular, it can generate consistent interaction sequences and explainable reward functions at interaction states, opening the path for training generative models of the future.

## 1 Introduction

Recent progress in generative artificial intelligence with models capable of generating creative content has shown attractive prospects for real-world applications, such as image generation (Takagi & Nishimoto, 2023), embodied agents (Brohan et al., 2023b), and chatbots (Köpf et al., 2024). Most generative models attempt to directly obtain the answer by training on natural language or image datasets and inserting decomposed reasoning steps in few-shot demonstrations. However, these methods do not experience firsthand the situations described by the language and the image. They

---

[*]Corresponding authors.

38th Conference on Neural Information Processing Systems (NeurIPS 2024).

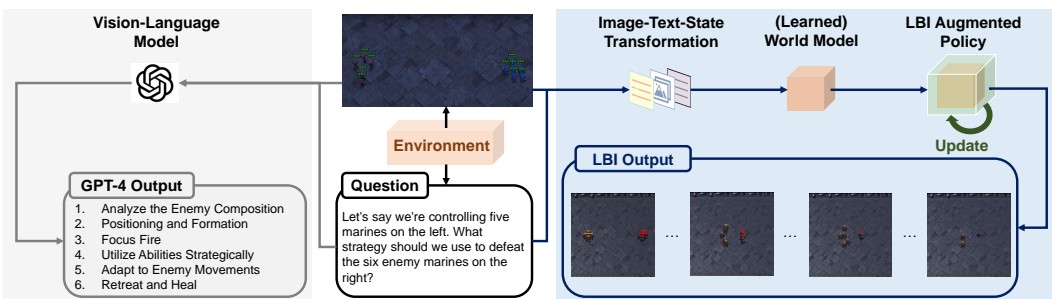

Figure 1: Complex decision problems that require a good understanding of the environment's dynamics and the objective are still challenging for current vision-language models, e.g., the answer elicited by GPT-4 is sketchy and misleading. Instead, Learning before Interaction (LBI) enables grounded reasoning by simulating the task in the given question. LBI utilizes the simulator to train a MARL policy and generate the answer by running the converged policy on the simulator.

cannot find the correct answers through trial and error as humans, which is necessary to ground reasoning on complicated problems and transfer learned knowledge to unfamiliar domains. For example, as shown in Figure 1, when asked a complex multi-agent decision problem, one of the most widely-used large language models, GPT4 - though achieving superhuman performance in many reasoning tasks - will generate sketchy and misleading answers.

To tackle this problem, we can utilize the generative models to understand the properties of the task that the user describes and simulate the effects of the actions. We can derive the answer with a highly realistic simulator by experiment-reasoning or training any machine intelligence from simulated experience. The origin of this idea can be traced back to Dyna architecture (Sutton, 1990) and has spawned a series of model-based reinforcement learning (MBRL) theories and methods (Janner et al., 2019; Kaiser et al., 2019; Lai et al., 2020). Inspired by this, Mind's Eye (Liu et al., 2022) enables language models to perform reasoning conditioned on the simulation results by running the corresponding experiment on a computational physics engine named MuJoCo (Todorov et al., 2012). Mind's Eye can boost reasoning performance in zero-shot and few-shot settings by infusing such physical knowledge into language models. However, it is particularly designed for physical reasoning rather than decision-making problems.

In contrast, UniSim (Yang et al., 2024) formulates the action-in-video-out framework as an observation prediction diffusion model conditioned on finite history. It shows that the simulator learned from broad data can generalize to the real world and bridge the sim-to-real gap. Genie (Bruce et al., 2024) enables users to act in the generated environments on a frame-by-frame basis, opening the path for training generalist agents of the future. Notably, most of the existing breakthroughs on learning in the imagined experience have been focusing on single-agent scenarios and leave the world model largely unstudied for multi-agent reinforcement learning (MARL) tasks - it is common in real-world applications that multiple agents are required to solve a task in a coordinated fashion.

The roadblocks to building a simulator for MARL tasks are twofold. First, MARL tasks involve multiple entities' attributes, e.g., positions and roles, making it difficult to describe a state using only text. The text and image information can be brought together to enrich the inputs for the simulator, but such a dataset is limited in quantity. Second, the dynamics and reward models of the MARL environment are more intricate than the single-agent setting. Current approaches assume the reward is known in the dataset (Meng et al., 2023) or can be easily deduced by the frame information (Yang et al., 2024), which could be challenging for MARL methods due to the abundance of agents' tactics and the compositional nature of their functionalities.

This work explores a paradigm that adds language-guided simulation to the MARL pipeline to make policy learning grounded within the learned world model. First, we propose new offline MARL datasets to provide paired state-image information for the StarCraft Multi-Agent Challenge (SMAC) environment by transforming the state in the trajectory to the corresponding image. We also designed a parser to convert each trajectory to a task description using natural language. Then, we pre-train a vector quantized variational autoencoder (VQ-VAE) (Van Den Oord et al., 2017) to generate discrete representations for each frame. The world model is formulated as an interactive simulator that consists

of a dynamics and a reward model. The dynamics model comprises an image tokenizer and a causal transformer to generate interaction transitions autoregressively. The reward model is a bidirectional transformer learned by maximizing the likelihood of trajectories in the expert demonstrations under the task description.

Given a decision-making problem by the user and an image from the environment, we store the simulated interaction trajectories into a replay buffer by running an off-policy MARL algorithm on the generated dynamics model. Then, we utilize the generated reward model to label the reward for each state-action pair based on the whole trajectory. We update the policy network according to the reward with a behavior-regularization term, which serves as the conservatism for out-of-distribution state-action pairs. We use the image sequence generated by the interaction of the dynamics model and the converged policy model as the answer to the decision-making problem.

We summarize the main contributions of this paper in three folds: (1) It proposes novel MARL datasets for SMAC, where a parser automatically generates the ground-truth image of a given state and task description. (2) It introduces Learning before Interaction (LBI), an interactive simulator that generates trial-and-error experiences and improves the answers for multi-agent decision-making problems. (3) The empirical results show that LBI outperforms various offline learning methods on training and unseen MARL tasks. The visualization also indicates that LBI can produce consistent imagined trajectories and explainable rewards for interaction states.

## 2  Background

**Decentralized Partially Observable Markov Decision Process.** A fully cooperative multi-agent task in the partially observable setting can be formulated as a Decentralized Partially Observable Markov Decision Process (Dec-POMDP) (Oliehoek & Amato, 2016), consisting of a tuple $G = \langle A, S, \Omega, O, U, P, r, \gamma \rangle$, where $a \in A \equiv \{1, \ldots, n\}$ is a set of agents, $S$ is a set of states, and $\Omega$ is a set of joint observations. At each time step, each agent obtains its observation $o \in \Omega$ based on the observation function $O(s, a) : S \times A \to \Omega$, and an action-observation history $\tau_a \in T \equiv (\Omega \times U)^*$. Each agent $a$ chooses an action $u_a \in U$ by a stochastic policy $\pi_a(u_a|\tau_a) : T \times U \to [0, 1]$, which forms a joint action $\mathbf{u}$. It results in a joint reward $r(s, \mathbf{u})$ and a transit to the next state $s' \sim P(\cdot|s, \mathbf{u})$. The formal objective function is to find the joint policy $\boldsymbol{\pi}$ that maximizes a joint action-value function $Q^{\boldsymbol{\pi}}(s_t, \mathbf{u}_t) = r(s_t, \mathbf{u}_t) + \gamma \mathbb{E}_{s'}[V^{\boldsymbol{\pi}}(s')]$, where $V^{\boldsymbol{\pi}}(s) = \mathbb{E}[\sum_{t=0}^{\infty} \gamma^t r_t | s_0 = s, \boldsymbol{\pi}]$, and $\gamma \in [0, 1)$ is a discounted factor.

**Inverse Reinforcement Learning.** Suppose we do not have access to the ground truth reward function but have demonstrations $\mathcal{D}$ provided by an expert policy $\pi_E$. Imitation learning aims to directly learn policies that behave similarly to these demonstrations, whereas inverse reinforcement learning (IRL) seeks to infer the underlying reward functions which induce the expert policies. The MaxEnt IRL framework (Ziebart et al., 2008) aims to recover a reward function that rationalizes the expert behaviors with the least commitment, denoted as IRL($\pi_E$):

$$
\begin{aligned}
\mathrm{IRL}(\pi_E) &= \arg\max_r \mathbb{E}_{\pi_E}[r(s, u)] - \mathrm{RL}(r) \\
\mathrm{RL}(r) &= \max_\pi \mathcal{H}(\pi) + \mathbb{E}_\pi[r(s, u)]
\end{aligned}
\tag{1}
$$

where $\mathcal{H}(\pi) = \mathbb{E}_\pi[-\log \pi(u|s)]$ is the entropy of current policy $\pi$. It looks for a reward function that assigns a high reward to the expert policy and a low reward to the current policy $\pi$ while searching for the best policy for the reward function in the inner loop.

## 3  Methodology

We formulate an interaction simulator as a transition prediction model that, given some state of the world and descriptions of the task, can take some actions as input and produce the consequence of the actions in the form of images, states, and rewards. In this paper, we consider building such simulators for a multi-agent decision-making environment named StarCraft Multi-Agent Challenge (SMAC) (Samvelyan et al., 2019), known for its rich environments and high control complexity. More information about SMAC can be found in Appendix C.

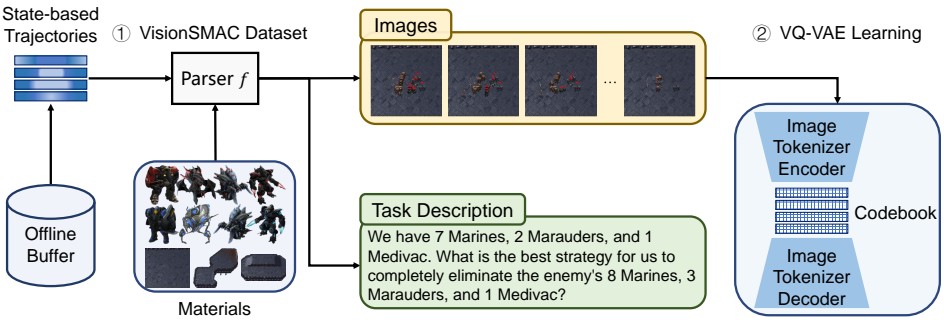

Figure 2: Datasets construction and VQ-VAE training.

### 3.1 VisionSMAC

The SMAC benchmark saves a replay of an episode as a SC2REPLAY file rather than providing the image feature during exploration. It is computationally expensive to construct datasets of images by watching such replay within the StarCraft II client and then subsampling a frame that captures meaningful actions. To solve this problem, we introduce VisionSMAC to convert the state into images and languages through a parser $f$, decoupled from StarCraft, making it easy to create new content.

First, we collect offline datasets across ten training maps in the SMAC benchmark by running multi-agent exploration methods named EMC (Zheng et al., 2021) and IIE (Liu et al., 2024). Each dataset contains a large number of interaction trajectories:

$$\tau := \{s_t, \{o_t^a\}_{a=1}^n, \{u_t^a\}_{a=1}^n, \{d_t^a\}_{a=1}^n\}_{t=0}^T \tag{2}$$

where $s_t$ denotes the state, $\{o_t^a\}_{a=1}^n$ is the observation of each agent, $\{u_t^a\}_{a=1}^n$ is the joint action, and the done signal $d_t^a = 1$ when the agent $a$ is killed at timestep $t$, $n$ and $T$ denote the number of agents and the length of the episode, respectively. We further collect the element images that appear in the game and affect the state, such as the units and the background terrain of training maps.

Given a multi-agent system and its interaction trajectory, the parser $f$ reads predefined map information, such as the number and races of agents and enemies. Then, the parser converts the original state information into structured information, reading agents' and enemies' positions and health points. It will generate the corresponding interaction scenes by placing each unit image and its health bar at their positions with the corresponding background terrain. The image generated by the parser can resemble the frame subsampled from the original replay by running a SC2REPLAY file within the StarCraft II client, where the comparisons can be found in Appendix B. We also perform data augmentation to enable better feature abstraction by changing the background to different terrains.

Finally, we define a task description $L$ to specify the environment and the task. The task description can be the terminated state, a slice of a trajectory, or any other representation of the episode. In this paper, we use the terrain information, the number and unit types of agents and enemies, and the sum of enemies' remaining health points at the terminated state as the task description. To this end, the parser reads the last state of the trajectory and extracts the remaining health points of both sides. We can obtain the practical task description by filling in predefined description templates (e.g., "*Consider that we control {number of agents} {agent races} on the left.*") and adding connecting words (e.g., "*What plan should we use*"). The detailed description of the datasets and the parser $f$ can be found in Appendix B.

### 3.2 Training An Interactive Simulator

With trajectories with corresponding images and language guidance from different scenarios, we can formulate the interactions with StarCraft II as interacting with an interactive simulator. The simulator contains three key components: (1) an image tokenizer that converts raw video frame into discrete tokens, (2) a dynamics model that predicts the next frame and state given past frame and state tokens, (3) a reward model that infers the reward of a state-action pair given a trajectory. The idea behind decomposing the world model into a dynamics model and a reward model is to reuse the dynamics model for different tasks by combining it with any reward function.

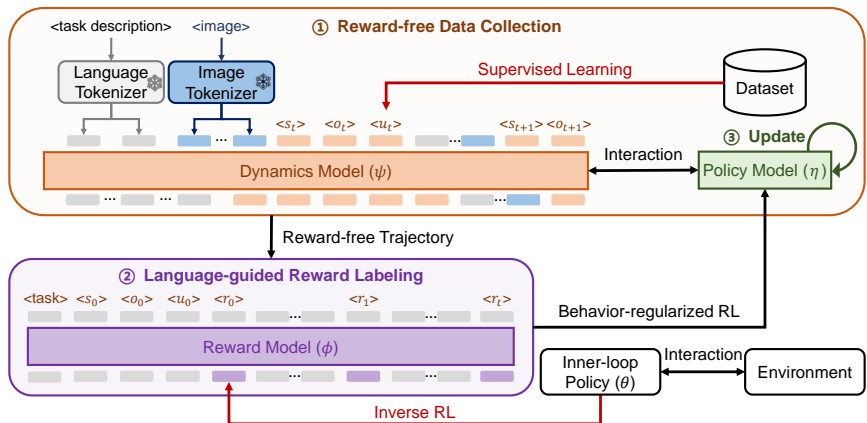

Figure 3: The overview of Learning before Interaction.

**Image Tokenizer**   We compress images into discrete tokens to reduce dimensionality and enable higher-quality image generation. We make use of vector quantized variational autoencoder (VQ-VAE) (Van Den Oord et al., 2017), which takes every single image of the state as input, generating discrete representations. The tokenizer is trained using a standard VQ-VAE objective.

**Dynamics Model**   The dynamics model is a causal transformer $q$ parameterized by $\psi$, where the target sequence has the following form $x = \{..., L, z_t, s_t, o_t^1, ..., o_t^n, u_t^1, ..., u_t^n, z_{t+1}, s_{t+1}, ...\}$, where $z_t$ is the image representation generated by the fixed image tokenizer. We utilize the task description $L$ to specify the dynamics of the environment, remaining consistent in one sequence. An embedding for each timestep is learned and added to each token. The dynamics model processes all past tokens and predicts future tokens via autoregressive modeling.

Then, we use the prediction heads to decode the predicted tokens to the corresponding element in the sequence and train them by minimizing the cross-entropy loss for actions and mean-squared error for others. The actions would serve as the reference policy for the learning with the simulated trajectories described in Section 3.3. In particular, we use a dynamics residual term to improve the accuracy and the stability of the generation by changing the target from $s_{t+1}$ to $\Delta s_{t+1} = s_{t+1} - s_t$ for the state prediction head. We also apply this term to predict image representations. In addition, since the observation is only related to the current state and the vision range of the agents, we filter out the historical memories and use $s_t$ as the input for the observation prediction.

**Reward Model**   We resemble the training pipeline of inverse reinforcement learning - maximizing the likelihood of trajectories in the expert demonstrations while minimizing the likelihood of trajectories collected from an inner-loop policy. We introduce a reward function $\hat{r}$, which receives the entire trajectory as inputs to perform credit assignment under deterministic dynamics within the trajectory. We remake this formulation as a generalized version of the conventional reward design; if the reward function is Markovian, the temporal dependencies on other state-action pairs should always be zero.

To this end, we model the reward function as a bidirectional transformer model parameterized by $\phi$, where the sequence is $\tilde{x} = \{..., s_t, L, u_t^1, ..., u_t^n, \hat{r}_t, s_{t+1}, ...\}$, and $\hat{r}_t = \{\hat{r}_t^a\}_{a=1}^N$ is a set of individual rewards for the agents. Again, we utilize the task description $L$ to perform hindsight relabeling, which converts imperfect trajectories into possible solutions for reaching the last state $s_T$ of the episode, generating the expert demonstration $\mathcal{D}$. We optimize the reward function by minimizing the following loss:

$$\nabla_\phi \mathcal{L} = -\mathbb{E}_{\tau \sim \mathcal{D}} \left[ \sum_i^N \sum_t \gamma^t \nabla_\phi \hat{r}_t^a(\tau; \phi) \right] + \mathbb{E}_{\tau \sim \boldsymbol{\pi}^\theta} \left[ \sum_i^N \sum_t \gamma^t \nabla_\phi \hat{r}_t^a(\tau; \phi) \right], \qquad (3)$$

where $\boldsymbol{\pi}^\theta = \{\pi^a(u^a|s; \theta)\}_{a=1}^N$ is the inner-loop MA-SAC policy parameterized by $\theta$, and we use the reward constraint by imposing an L2 penalization of the predicted rewards over all possible actions, which can be viewed as a conservative update for out-of-distribution state-action pairs. In particular, we alternate between $k$-step of policy update and $k$-step of reward update to avoid completely solving

the policy optimization subproblem before updating the reward parameters, where $k$ is a given update iteration.

### 3.3 Inference: Learning Policy in the Simulator

We now describe how to generate grounded answers for multi-agent decision-making problems via LBI. Given an image of the initial state and a task description from the user, the agents interact with the dynamics model using a randomly initialized off-policy MARL algorithm to collect reward-free trajectories in an autoregressive manner. Then, the reward model predicts the immediate reward for each transition pair in the simulation trajectories. These relabeled trajectories are added to the replay buffer, serving as the training data for the policy network.

In practice, we construct the MARL problem in the simulator as a behavior-regularized MDP by imposing a behavior-regularization term:

$$\max_{\bar{\boldsymbol{\pi}}} \mathbb{E}\left[ \sum_{t=0}^{\infty} \gamma \left( \sum_{a=1}^{n} \left( r_t^a(\tau; \phi) - \alpha \cdot \log\left( \frac{\bar{\pi}_i(u_t^a | o_t^a; \eta)}{q(u_t^a | x_{<u_t^a}; \psi)} \right) \right) \right) \right], \tag{4}$$

where $\bar{\boldsymbol{\pi}} = \{\bar{\pi}_i(u_t^a | o_t^a; \eta)\}_{a=1}^{n}$ is the joint policy, and $q(u_t^a | x_{<u_t^a}; \psi)$ is the reference policy provided by the dynamics model. The last term enables in-sample learning and further mitigate the impact of exploring OOD regions in the state-action space. We use independent $Q$-learning to train the parameter-sharing agent network.

Since it is possible for specific agents to become inactive before the game terminates, we mark the terminated timestep for each agent and enemy once its predicted health is less than zero and then use zero vectors as the subsequent actions and observations. It can mitigate the hallucinating unrealistic outcomes - a dead agent performs a "moving" action. We also mask the predicted reward after the terminated timestep for the inactive agent to get a more accurate value estimate.

## 4 Related Work

**World Models.** There is a long-standing history of learning predictive models of the world. We list three categories of model-based reinforcement learning (MBRL) according to the type of model usage. The first category applies planning methods with world model simulation. AlphaGo (Silver et al., 2016) and MuZero (Schrittwieser et al., 2020) learn a transition model and apply Monte Carlo Tree Search to search for an action sequence with the highest accumulated rewards. By contrast, MBMF (Nagabandi et al., 2018), PETS (Chua et al., 2018), and PlaNet (Hafner et al., 2019b) integrate model predictive control (MPC) into the learned world model and sample high-reward action sequences. TD-MPC (Hansen et al., 2022) and TD-MPC2 (Hansen et al., 2024) utilize value functions to bootstrap the trajectories used for MPC planning.

The second category models a differentiable world model and utilizes the internal structure of the model to facilitate policy learning. GPS (Levine & Koltun, 2013) and GDP (Srinivas et al., 2018) perform differential planning and obtain the analytic form of the optimal policy. SVG (Heess et al., 2015) re-parameterizes the policy and the world model, then computes the policy gradient estimate by backpropagation via the world model. MAAC (Clavera et al., 2019), Dreamer (Hafner et al., 2019a) and its subsequent variants (Hafner et al., 2020, 2023) use the recurrent state-space model in PlaNet to learn the world model in a compact latent space and learn the policy entirely within this space.

The last one utilizes the learned world model to generate more experiences and then trains a policy on the dataset augmented by the model, also known as Dyna-style methods (Sutton, 1990). MVE (Feinberg et al., 2018) and STEVE (Buckman et al., 2018) depict a learned world model to calculate multi-step temporal-difference prediction for better value estimation. In contrast, SLBO (Luo et al., 2018), MBPO (Janner et al., 2019), and BMPO (Lai et al., 2020) theoretically analyze this learning framework and prove that the policy performance will improve monotonically in a world model with certain model bias and rollout length. To further increase the rollout length and avoid compounding errors, M2AC (Pan et al., 2020) and COPlanner (Wang et al., 2023) compute the uncertainty of each rollout step and perform conservative model rollouts by discarding the samples with high uncertainty or adding a penalty term into total reward. In practice, GAIA-1 (Hu et al., 2023), UniSim (Yang et al., 2024), and Genie (Bruce et al., 2024) show that the learned world model can enable the control policy to generalize to the real world when trained purely in simulation and bridge the sim-to-real

gap. These methods have impressive performance and theoretical bounds, attracting much research interest in the MBRL community. However, they focus on generating videos or trajectories that only involve one single agent instead of building a multi-agent simulator that can be used to further improve decision-making performance on coordination tasks in our work.

**Imitation Learning.** Imitation Learning (Bain & Sammut, 1995) formulates imitating an expert as a supervised learning problem, which has been widely adopted in various domains due to its simplicity and effectiveness (Silver et al., 2016; Swamy et al., 2020). GAIL (Ho & Ermon, 2016) and its extensions (Song et al., 2018; Ghasemipour et al., 2020) stand as a cornerstone approach, which trains a generator policy to imitate expert behaviors and a discriminator to distinguish between the expert and the learner's state-action pair distributions. In light of the recent interest in foundational models, the conditional diffusion model is used to represent and learn an imitation learning policy, which produces a predicted action conditioning on a state and a sampled noise vector Pearce et al. (2022); Chi et al. (2023). These methods achieve encouraging results in modeling stochastic and multimodal behaviors from human experts or play data. DT-style methods (Chen et al., 2021; Wu et al., 2024) formulate the trajectory generation as a sequence modeling problem, which generates states, actions, and rewards by conditioning on a return-to-go token in an autoregressive manner.

In contrast, inverse reinforcement learning (IRL) is designed to infer the reward function that underlies the expert demonstrations, taking into account the temporal structure and showing better generalization than direct Behavioral Cloning (Ng & Russell, 2000; Ross et al., 2011; Barde et al., 2020). A main class of algorithms, Maximum entropy (MaxEnt) IRL (Haarnoja et al., 2017) and its extensions (Liu et al., 2021; Rolland et al., 2022), learn a stationary reward by minimizing divergence between the agent and expert distribution. Since the learned reward function can solve downstream tasks and transfer behavior across different dynamics, IRL is also helpful in several broader applications, e.g., IRL with natural language goals (Fu et al., 2018a; Zhou & Small, 2021; Xu et al., 2022), and RL with human feedback (Ziegler et al., 2019; Zhu et al., 2023; Wu et al., 2023), and dynamics learning (Luo et al., 2023). Furthermore, a series of sample-efficient algorithms are proposed to solve the MaxEnt IRL formulation (Fu et al., 2018b; Zeng et al., 2022, 2024). To side-step the expensive online environmental interactions in classic IRL, some work aims to learn a reward function from a static dataset by a variational Bayesian framework (Chan & van der Schaar, 2021), representing reward function via a learned soft $Q$-function (Garg et al., 2021), or incorporating conservatism into the estimated reward like offline $Q$-learning (Yue et al., 2022). The major bottleneck for these methods includes a lack of knowledge of the dynamics information and the reward overestimation for out-of-distribution state-action pairs. We formulate the reward model as a bidirectional transformer to receive the whole trajectory as the input, making it possible to solve non-Markovian rewards. In addition, we leverage the reward constraint and the behavior regularization to perform in-sample learning to avoid reward overestimation. The amount of expert demonstrations in these existing studies is also limited, as they do not treat hindsight relabeling via the textual description as an expert trajectory like in our work.

**Offline $Q$-Learning.** Offline $Q$-learning learns a policy from a fixed dataset where the reward is provided for each transition sample. Most off-policy reinforcement learning (RL) algorithms are applicable in offline $Q$-learning. However, they typically suffer from the overestimation problem of out-of-distribution (OOD) actions due to the distribution shift between the action distribution in the training dataset and that induced by the learned policy (Fujimoto et al., 2019). Several constraint methods are proposed to restrict the learned policy from producing OOD actions by leveraging importance sampling (Sutton et al., 2016; Nachum et al., 2019), incorporating explicit policy constraints (Kostrikov et al., 2021; Fakoor et al., 2021; Fujimoto & Gu, 2021; Tarasov et al., 2024), penalizing value estimates (Kumar et al., 2020; An et al., 2021; Shao et al., 2024), and uncertainty quantification (Wu et al., 2021; Zanette et al., 2021). Another branch resorts to learning without querying OOD actions and thus constrain the learning process within the support of the dataset (Bai et al., 2021; Lyu et al., 2022).

## 5    Experiments

In this section, we conduct empirical experiments to answer the following questions: (1) Is Learning before Interaction (LBI) better than the existing multi-agent reinforcement learning (MARL) methods in complex cooperative scenarios? (2) Can LBI generate long-horizon trajectories and reasonable reward functions at critical states? (3) Does LBI have the zero-shot ability to generalize to unseen

Table 1: Test win rates (%) and standard deviations compared with imitation learning methods.

| Map Name | BC | MA-AIRL | MADT | MAPT | MA-TREX | LBI |
|---|---|---|---|---|---|---|
| 1c3s5z | 16.44± 1.35 | 7.88± 2.49 | 61.35± 7.26 | 74.77± 5.15 | 64.76± 11.62 | 94.59± 3.41 |
| 10m_vs_11m | 26.19± 4.42 | 41.69± 7.12 | 82.76± 4.41 | 66.85± 9.28 | 48.78± 11.28 | 90.45± 6.99 |
| 2c_vs_64zg | 17.37± 10.12 | 24.75± 10.83 | 61.90± 5.74 | 58.28± 7.84 | 22.45± 7.74 | 71.44± 8.83 |
| 3s_vs_5z | 0.00± 0.00 | 0.05± 0.03 | 80.90± 0.45 | 72.33± 3.93 | 55.38± 18.03 | 92.82± 6.25 |
| 5m_vs_6m | 13.78± 2.15 | 11.59± 6.75 | 79.78± 4.98 | 56.01± 3.17 | 50.01± 14.87 | 87.98± 5.10 |
| 6h_vs_8z | 9.28± 5.06 | 16.47± 8.08 | 30.94± 25.54 | 37.16± 6.27 | 28.38± 5.31 | 66.61± 4.57 |
| 3s5z_vs_3s6z | 0.00± 0.00 | 0.00± 0.00 | 27.44± 9.49 | 34.90± 6.84 | 36.16± 3.68 | 83.34± 4.27 |
| corridor | 0.00± 0.00 | 0.76± 0.15 | 69.85± 1.54 | 45.91± 15.47 | 30.59± 9.86 | 87.45± 2.94 |
| MMM2 | 0.00± 0.00 | 0.00± 0.00 | 54.34± 12.83 | 19.21± 5.59 | 21.52± 6.58 | 95.96± 4.65 |

Table 2: Test return and standard deviations compared with offline reinforcement learning methods.

| Map Name | BCQ-MA | CQL-MA | ICQ | OMAR | OMIGA | LBI |
|---|---|---|---|---|---|---|
| 5m_vs_6m | 9.13± 0.21 | 10.15± 0.15 | 9.47± 0.45 | 8.76± 0.52 | 10.38± 0.50 | 18.96± 0.56 |
| 2c_vs_64zg | 18.86± 0.35 | 19.20± 1.25 | 18.47± 0.25 | 17.10± 0.94 | 19.25± 0.38 | 20.45± 0.25 |
| 6h_vs_8z | 11.91± 0.44 | 9.95± 0.32 | 11.55± 0.15 | 9.74± 0.28 | 12.74± 0.21 | 18.97± 0.28 |
| corridor | 16.42± 1.55 | 6.64± 0.90 | 16.74± 1.78 | 8.15± 0.89 | 17.10± 1.33 | 19.50± 0.73 |

tasks? Then, we investigate the contribution of each component in the dynamics and the reward model. We provide the information of training datasets and experimental settings in Appendix B and D. We also discuss this paper's broader impacts and limitations in Appendix A.1 and A.2.

## 5.1 Performance Comparison

**Reward-free Offline Learning** We compare LBI with the following imitation learning baselines: (1) BC: behavior cloning that imitates the whole datasets, (2) MA-AIRL (Yu et al., 2019): using adversarial learning to perform policy imitation, (3) MADT (Meng et al., 2023): utilizing the Decision Transformer (Chen et al., 2021) to perform sequence modeling, (4) MA-TREX: infering the reward according to ranked demonstrations, the multi-agent version of TREX (Brown et al., 2019), (5) MAPT (Zhu et al., 2024): infering the team rewards according to the preference return from a well-trained scripted teacher.

As shown in Table 1, LBI outperforms the baselines by a significant margin on various maps with different difficulty levels, indicating the importance and effectiveness of learning reward functions via the proposed world model. In contrast, BC and MA-AIRL fail to achieve success rates in most tasks because they imitate all past interaction sequences and cannot generalize and avoid sub-optimal solutions. MA-TREX and MAPT have plateaued in performance because they use the accumulated rewards and the preference deduced by the scripted teacher to specify the quality of the training data, respectively. MADT performs better than other baselines because Decision Transformer can be thought of as performing imitation learning on a subset of the data with a certain return.

**Offline MARL** We also compare LBI with the existing offline MARL methods with ground-truth rewards from the StarCraft Multi-Agent Challenge (SMAC), including the multi-agent version of BCQ (Fujimoto et al., 2019) and CQL (Kumar et al., 2020) (namely BCQ-MA and CQL-MA), ICQ (Yang et al., 2021), OMAR (Pan et al., 2022), and OMIGA (Wang et al., 2024). Table 2 shows that the performance of these offline MARL methods degrades dramatically with an increasing number of agents and is much lower than that of LBI. We hypothesize that the reasons for this gap are: (1) It is challenging and unnecessary to recover the $Q$-value based on the reward functions provided by SMAC (the hit-point damage dealt) because such reward design is inefficient for learning optimal policy. (2) These methods may introduce too much conservatism and affect the learning of the optimal policy, as the conservative update of the out-of-distribution (OOD) suboptimal policy that consists of some agents taking non-optimal actions and others taking optimal will inhibit the learning of the agents that take the optimal actions.

## 5.2 Generalization on Unseen Tasks

Since zero-shot generalization ability is crucial for generating grounded answers for multi-agent decision-making problems, we also test LBI's ability to generalize to extensive unseen scenarios

Table 3: Test win rates (%) and standard deviations on unseen tasks.

| Unseen Task | MADT | MA-TREX | LBI | Unseen Task | MADT | MA-TREX | LBI |
|---|---|---|---|---|---|---|---|
| 1c3s | 16.21± 5.38 | 23.53 ± 8.83 | 56.47± 5.63 | 1c2s7z | 6.16± 3.09 | 5.69±3.81 | 28.26± 6.41 |
| 6m | 49.28± 4.06 | 37.12±2.59 | 97.85± 2.15 | 6m_vs_7m | 73.45± 7.22 | 32.88±4.47 | 81.07± 5.17 |
| 1c_vs_32zg | 2.08± 1.51 | 11.41±3.41 | 58.33± 6.44 | 3s4z | 90.21± 1.82 | 79.71±3.56 | 87.55± 1.76 |
| 3s2z_vs_2s3z | 0.00± 0.00 | 9.16±5.62 | 18.22± 2.46 | 3s5z_vs_3s7z | 10.21± 3.66 | 15.88±4.34 | 22.08± 7.63 |
| 1c3s6z | 16.41± 6.44 | 58.09±3.41 | 65.38± 5.12 | 9m_vs_11m | 76.44± 4.17 | 70.91±6.95 | 75.05± 2.16 |

Table 4: The ablation results for the dynamics model without residual term (wo-RT), image reference (wo-IR), and using ground-truth image (GTI) as the reference for state prediction.

Table 5: The ablation results for the reward model without reward constraint (wo-RC), behavior regularization (wo-BR), and using ground-truth rewards (w-GTR) provided by the SMAC benchmark.

| Algorithm | Prediction Error | Return (All) |
|---|---|---|
| LBI | 0.016 ± 0.023 | 18.91 ± 1.33 |
| LBI-GTI | 0.014 ± 0.018 | 18.98 ± 0.89 |
| LBI-wo-RT | 0.434 ± 0.351 | 14.25 ± 1.84 |
| LBI-wo-IR | 0.029 ± 0.041 | 18.63 ± 1.01 |
| LBI-wo-RT&IR | 0.744 ± 1.164 | 12.13 ± 2.33 |

| Algorithm | Return (Training) | Return (unseen) |
|---|---|---|
| LBI | 19.47± 0.77 | 18.54 ± 1.49 |
| LBI-GTR | 16.68 ± 1.55 | 14.07 ± 2.79 |
| LBI-wo-RC | 17.85 ± 0.59 | 14.75 ± 1.67 |
| LBI-wo-BR | 18.82 ± 1.28 | 17.46 ± 2.01 |
| LBI-wo-RC&BR | 12.35 ± 2.38 | 9.83 ± 1.46 |

without retraining. Specifically, we evaluate our LBI and MADT on the ten unseen testing maps, varying agent numbers, action spaces, and levels of environment complexity. Table 3 shows that LBI consistently outperforms MADT in unseen scenarios by a large margin, successfully transferring knowledge to new tasks without requiring additional fine-tuning. It highlights that learning a reward function has better zero-shot generalization performance than simple policy adaptation.

## 5.3 Ablation Studies

In this section, we conduct ablation studies to analyze the contributions of each component in the dynamics model and the reward model across five evaluation runs on four training maps (6h_vs_8z, 3s5z_vs_3s6z, corridor, and MMM2) and four unseen maps (3s5z_vs_3s7z, 1c3s7z, 3s4z, 1c_vs_32zg). We show the results of the dynamics model in Table 4. Using the dynamics residual term is necessary to reduce the prediction error of the subsequent states and obtain good performance across all training and unseen tasks. The image reference is not so effective, even if we use ground-truth images as the reference. However, since images are more powerful in representing some situations than language or state information, we believe that the image serves as another modality to correct the prediction of the state. We would leave it for future work.

We demonstrate the ablation results of the reward model in Table 5. Compared with LBI-wo-RC&BR, the reward constraint and behavior regularization term can improve the overall performance on the training tasks. However, LBI-wo-BR performs better than LBI-wo-RC on unseen tasks, suggesting that the conservatism for reward is more important than the policy when OOD state-action pairs exist. The poor performance of LBI-w-GTR indicates that learning rewards from conditioned demonstrations may be more accessible and valuable for policy updates than reconstructing the pre-defined rewards by human experts.

## 5.4 Visualization

This section evaluates the dynamics model as a long-horizon policy-conditioned predictive model. Figure 4 showcases examples of length-40 image trajectories generated by the dynamics model, including MMM2, 3s_vs_5z, and 5m_vs_6m. We do not observe conspicuous compounding errors as the single-step prediction model does, highlighting that LBI has consistency and long-horizon generation ability. In the case of 5m_vs_6m, we present the following frames after taking one of the possible actions, showing that LBI can also perform action-controllable generation.

We also investigate the reward prediction at a critical junction in the state-action space that can transit to various states and significantly influence the success rate on the 5m_vs_6m task. At the moment, the agents have to learn to micromanage leapfrogging to achieve good coordination. Specifically, Agent 1 has a low health point and must move backward to avoid the enemies focusing fire on it; otherwise, the enemies will eliminate Agent 1 immediately and weaken our scarce forces. In Figure 4,

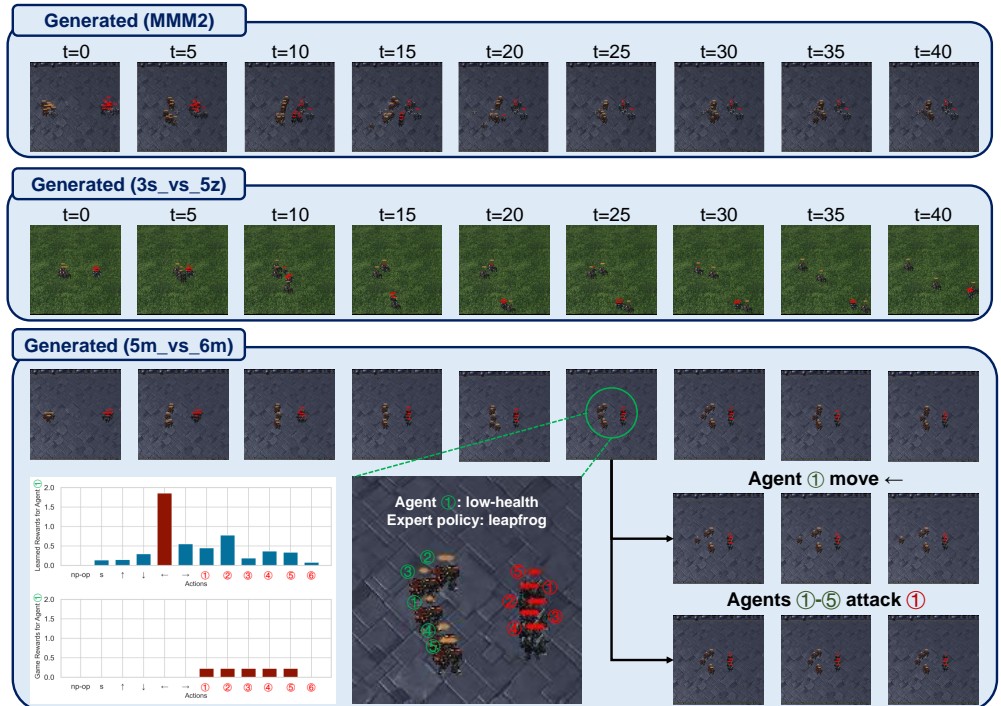

Figure 4: Visualization of the prediction from dynamics and reward model, where "np-op" and "s" denote no-operation and stopping, respectively.

we visualize the learned reward function of Agent 1, where the action space is no-operation, stopping, moving in cardinal directions, and selecting an enemy's identity to attack. The learned reward for moving to the left is much higher than the other actions, allowing one to learn the optimal joint policy quickly. The rewards provided by the SMAC benchmark do not show this property, where multiple Monte Carlo samples are required to find the correct policy by estimating the expected return.

## 6    Conclusion and Future Work

We proposed Learning before Interaction (LBI), a novel paradigm that enables generative models to ground their answers for multi-agent decision-making problems with simulations between the world and the multi-agent system. We formulate an interactive simulator consisting of dynamics and reward models, given some states of the world and the task descriptions, generating the consequence of the actions in the form of images, states, and rewards. We hope the idea of including simulations in the reasoning will instigate broad interest in applying generative models to aid machine intelligence and decision-making.

## 7    Acknowledgements

This work was supported in part by NSFC under grant No.62125305, No.62088102, No. U23A20339, No. 62203348.

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

# A  Broader Impacts and Limitations

## A.1  Broader Impacts

Learning before Interaction provides grounded answers to complex multi-agent decision-making problems through the generation of simulators and trial-and-error learning. This can benefit those seeking to make decisions through long-term planning. With significant technological advancements, exploring the use of this technology may be crucial for enhancing existing human decision-making capabilities. For instance, negotiators could describe the opponent's personality traits and their decision-making limits to generate better negotiation strategies.

At the same time, we recognize that current generative simulators still cannot reliably generate state transitions across multiple domains, and learning joint multi-agent strategies still faces convergence difficulties. Therefore, Learning before Interaction may lead to incorrect decisions in specific fields. If humans intentionally follow the generated answers instead of using them as references, it could lead to unsafe or worse consequences. On the other hand, it could also have negative impacts when Learning before Interaction is misused in harmful applications if the generated environments and answers are sufficiently accurate.

## A.2  Limitations

Although we have already seen significant improvements in reasoning capabilities for complex multi-agent tasks with Learning before Interaction, performance may be affected by the simulator's accuracy and the multi-agent policy learning performance. Unqualified simulators and difficult-to-converge multi-agent policies may lead to erroneous simulation results, which could be more misleading than the vague answers generated by existing visual language models. For example, the world model has limited out-of-domain generalization for domains that are not represented in the training data, e.g., unseen unit types. Further scaling up training data could help, as the parser can quickly and automatically generate images based on a given state.

While the learned reward functions can enhance the speed of multi-agent policy learning compared to other inverse reinforcement learning and online interaction learning methods, it still requires considerable waiting time to obtain a converged policy and the final answer. Such long waiting time is unacceptable in applications requiring real-time feedback, such as chatbots. One possible solution is to replace multi-agent reinforcement learning with planning methods based on the learned rewards and dynamics models, thereby accelerating the reasoning process. We will leave this issue in future work.

In addition, this paper is confined to scenarios within the game StarCraft II. This is an environment that, while complex, cannot represent the dynamics of all multi-agent tasks. Evaluation of multi-agent reinforcement learning algorithms, therefore, should not be limited to one benchmark but should target a variety with a range of tasks.

| Map Name | Return Distribution | Map Name | Return Distribution |
|---|---|---|---|
| 3s5z | $19.43 \pm 1.86$ | 5m_vs_6m | $19.83 \pm 2.16$ |
| 1c3s5z | $19.66 \pm 1.25$ | 6h_vs_8z | $18.84 \pm 2.09$ |
| 10m_vs_11m | $19.75 \pm 1.03$ | 3s5z_vs_3s6z | $19.76 \pm 1.26$ |
| 2c_vs_64zg | $19.98 \pm 0.71$ | corridor | $19.69 \pm 1.48$ |
| 3s_vs_5z | $19.88 \pm 1.40$ | MMM2 | $19.63 \pm 2.07$ |

Table 6: Return distribution on training maps.

# B  Dataset Preparation

The training maps include 3s5z, 1c3s5z, 10m_vs_11m, 2c_vs_64zg, 3s_vs_5z, 5m_vs_6m, 6h_vs_8z, 3s5z_vs_3s6z, corridor, MMM2 in StarCraft Multi-Agent Challenge (SMAC) (Samvelyan et al., 2019). We use EMC (Zheng et al., 2021) and IIE (Liu et al., 2024) to collect 50000 trajectories for each map and save these data as NPY files. The data includes the states, the observations, the

terminated signals, the actions, the available actions, and the rewards. The return distribution on training maps is shown in Table 6. The average return is $19.64 \pm 1.63$ across ten training maps.

As shown in Figure 5, we collect the element images that appear in the game and affect the state, including units and background terrains of training maps. In Figure 6, we have presented the whole procedure of converting a state vector into an image for simulation and parsing a trajectory to produce a textual task description.

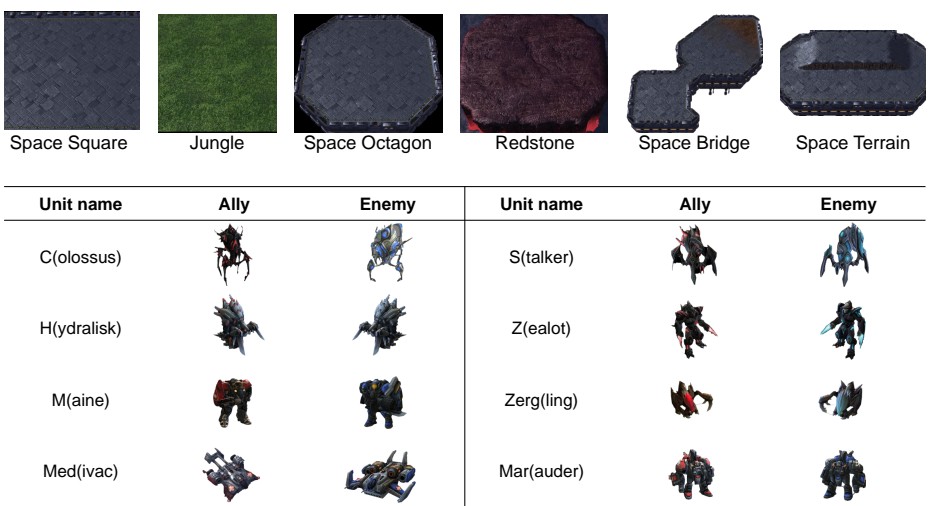

| Unit name | Ally | Enemy | Unit name | Ally | Enemy |
|-----------|------|-------|-----------|------|-------|
| C(olossus) | | | S(talker) | | |
| H(ydralisk) | | | Z(ealot) | | |
| M(aine) | | | Zerg(ling) | | |
| Med(ivac) | | | Mar(auder) | | |

Figure 5: Images of units and terrains.

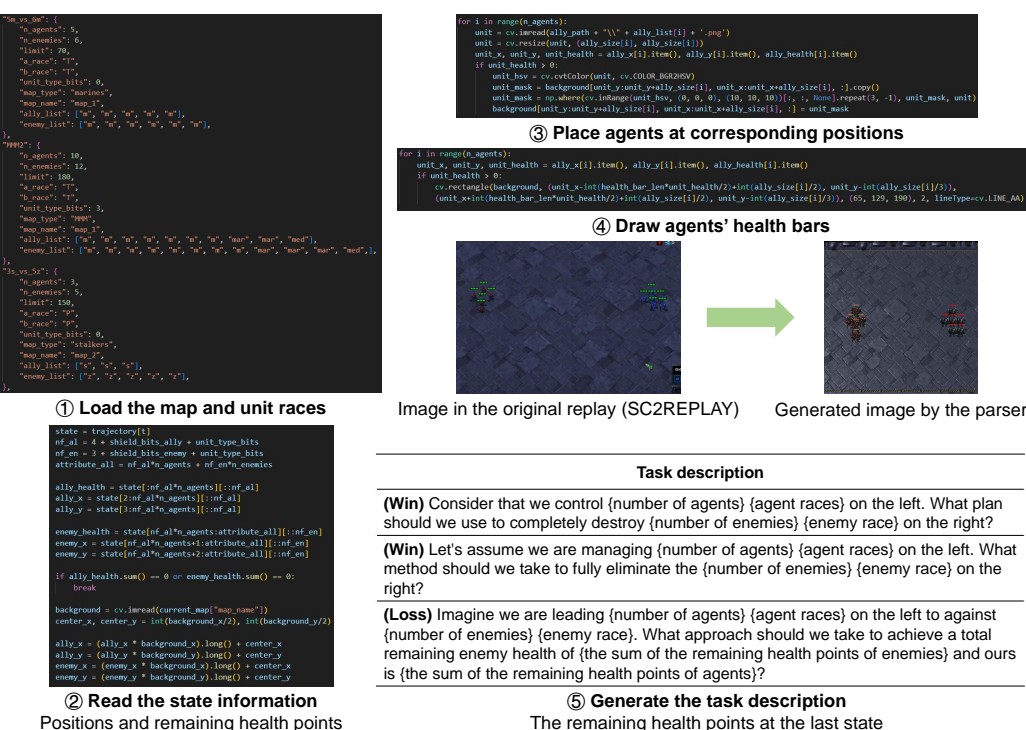

Figure 6: The whole pipeline of how the parser generates the image and the task description for a given state. Here, we only show three task descriptions the parser produces for demo purposes.

## C  StarCraft Multi-agent Challenge

StarCraft II is a real-time strategy game featuring three different races, Protoss, Terran, and Zerg, with different properties and associated strategies. The objective is to build an army powerful enough to destroy the enemy's base. When battling two armies, players must ensure army units are acting optimally. StarCraft Multi-Agent Challenge (SMAC) (Samvelyan et al., 2019) is a partially observable reinforcement learning benchmark built in StarCraft II. An individual agent with parameter sharing controls each allied unit, and a hand-coded built-in StarCraft II AI controls enemy units. The difficulty of the game AI is set to the "very difficult" level.

On the SMAC benchmark, agents can access their local observations within the field of view at each time step. The feature vector contains attributes of both allied and enemy units: `distance`, `relative x`, `relative y`, `health`, `shield`, and `unit_type`. In addition, agents can observe the last actions of allied units and the terrain features surrounding them. The global state vector includes the coordinates of all agents relative to the center of the map and other features present in the local observation of agents. The state stores the energy of Medivacs, the cooldown of the rest of the allied units, and the last actions of all agents. Note that the global state information is only available to agents during centralized training. All features in state and local observations are normalized by their maximum values. After receiving the observations, each agent is allowed to take action from a discrete set which consists of `move[direction]`, `attack[enemy_id]`, `stop` and `no-op`. Move direction includes north, south, east, and west. Note that the dead agents can only take `no-op` action while live agents cannot. For health units, Medivacs use `heal[agent_id]` actions instead of `attack[enemy_id]`.

Depending on different scenarios, the maximum number of actions varies between 7 and 70. Note that agents can only perform the `attack[enemy_id]` action when the enemy is within its shooting range. At each time step, agents take joint action and receive a positive global reward based on the total damage dealt to the enemy units. In addition, they can receive an extra reward of 10 points after killing each enemy unit and 200 points after killing all enemy units. The rewards are scaled to around 20, so the maximum cumulative reward is achievable in each scenario.

## D  Experiment Setting

In this section, we describe the ground-truth environment that agents interact, the implementation details of online learning methods, offline learning methods, and our model Learning before Interaction.

### D.1  Online Learning

We adopt the same architectures for QMIX, QPLEX, CW-QMIX[1], RODE[2], MAVEN[3], EMC[4] as their official implementations (Samvelyan et al., 2019; Wang et al., 2020a; Rashid et al., 2020; Wang et al., 2020c; Mahajan et al., 2019; Zheng et al., 2021). Each agent independently learns a policy with fully shared parameters between all policies. We used RMSProp with a learning rate of 5e-4 and $\gamma = 0.99$, buffer size 5000, and mini-batch size 32 for all algorithms. The dimension of each agent's GRU hidden state is set to 64.

For our experiments, we employ an $\epsilon$-greedy exploration scheme for the joint policy, where $\epsilon$ decreases from 1 to 0.05 over 1 million timesteps in `6h_vs_8z`, `3s5z_vs_3s6z` and `corridor`, and over 50 thousand timesteps in other maps. The implementation of MAPPO is consistent with their official repositories[5] (Yu et al., 2022). As shown in Table 7, all hyperparameters are left unchanged at the origin best-performing status. For CW-QMIX, the weight for negative samples is set to $\alpha = 0.5$ for all scenarios.

---

[1]https://github.com/oxwhirl/wqmix
[2]https://github.com/TonghanWang/RODE
[3]https://github.com/AnujMahajanOxf/MAVEN
[4]https://github.com/kikojay/EMC
[5]https://github.com/zoeyuchao/mappo

| Hyperparameter | Value | Hyperparameter | Value |
|---|---|---|---|
| critic lr | 5e-4 | actor lr | 5e-4 |
| ppo epoch | 5 | ppo-clip | 0.2 |
| optimizer | Adam | batch size | 3200 |
| optim eps | 1e-5 | hidden layer | 1 |
| gain | 0.01 | training threads | 32 |
| rollout threads | 8 | $\gamma$ | 0.99 |
| hidden layer dim | 64 | activation | ReLU |

Table 7: Hyper-parameters in MAPPO.

All figures in online learning experiments are plotted using mean and standard deviation with confidence internal 95%. We conduct five independent runs with different random seeds for each learning curve.

## D.2 Offline Learning

We adopt the same architectures for MA-AIRL[6], MADT[7], MAPT[8], ICQ[9], OMAR[10], and OMIGA[11] as their official implementations (Yu et al., 2019; Meng et al., 2023; Zhu et al., 2024; Fujimoto et al., 2019; Kumar et al., 2020; Yang et al., 2021; Pan et al., 2022; Wang et al., 2024). We implement MA-TREX, BCQ-MA and CQL-MA based on TREX (Brown et al., 2019), BCQ (Fujimoto et al., 2019), and CQL (Kumar et al., 2020), respectively. In particular, we add the task description into MADT's target sequence because it deprecates the reward-to-go term.

## D.3 Learning before Interaction

We train our image tokenizer for 100k steps using the AdamW optimizer, with cosine decay, using the hyperparameters in Table 8. The batch size is 32, and the learning rate is 1e-4.

| Component | Hyperparameter | Value |
|---|---|---|
| Encoder | num_layers | 5e-4 |
| | num_res_layers | 2 |
| | num_channels | (256,256) |
| | num_res_channels | (256,256) |
| | downsample | (2,4,1,1) |
| Decoder | num_layers | 5e-4 |
| | num_res_layers | 2 |
| | num_channels | (256,256) |
| | num_res_channels | (256,256) |
| | upsample | (2,4,1,1,0) |
| Codebook | num_codes | 256 |
| | latent_dim | 32 |
| | commitment_cost | 0.25 |

Table 8: Hyper-parameters in VQ-VAE.

We build our dynamics model implementation based on Decision Transformer[12] (Chen et al., 2021). The complete list of hyperparameters can be found in Table 9. The dynamics models were trained using the AdamW optimizer.

---

[6]https://github.com/ermongroup/MA-AIRL

[7]https://github.com/ReinholdM/Offline-Pre-trained-Multi-Agent-Decision-Transformer

[8]https://github.com/catezi/MAPT

[9]https://github.com/YiqinYang/ICQ

[10]https://github.com/ling-pan/OMAR

[11]https://github.com/ZhengYinan-AIR/OMIGA

[12]https://github.com/kzl/decision-transformer

| Hyperparameter | Value | Hyperparameter | Value |
|---|---|---|---|
| number of layers | 6 | grad norm clip | 1.0 |
| attention heads | 8 | weight decay | 0.1 |
| embedding dims | 64 | Adam betas | (0.9,0.95) |

Table 9: Hyperparameters in the transformer model.

The reward shares the same architecture as the dynamics model, but the attention mask in the transformer model is modified in order to receive the whole trajectory as input rather than the tokens that have come before the current one. Here are some tricks for reward learning: (1) we control the gap between the rewards of the expert behavior and the policy action - we stop the gradient for the reward of the expert behavior at a given state if it is greater than the one of the policy action, where $beta$ is the margin and set to 2; (2) we also set the target of unavailable actions' rewards to 0; (3) we alternate between $k$-step of policy update and reward update to avoid completely solving the policy optimization subproblem before updating the reward parameters, where $k = 5$.

In this paper, all experiments are implemented with Pytorch and executed on eight NVIDIA A800 GPUs.

# E  Additional Results

Using a Text-to-Code Converter can generate scenarios with the original game engine and then learn the joint policy. Therefore, we also consider the comparison with online MARL methods including CW-QMIX (Rashid et al., 2020), QPLEX (Wang et al., 2020a), MAVEN (Mahajan et al., 2019), EMC (Zheng et al., 2021), RODE (Wang et al., 2020c), QMIX (Rashid et al., 2018), MAPPO (Yu et al., 2022). Figure 7 demonstrates a significant improvement in the sample efficiency of LBI compared to the online MARL methods, suggesting that a pre-trained world model is necessary to reduce the waiting time for generating grounded answers for multi-agent decision-making problems.

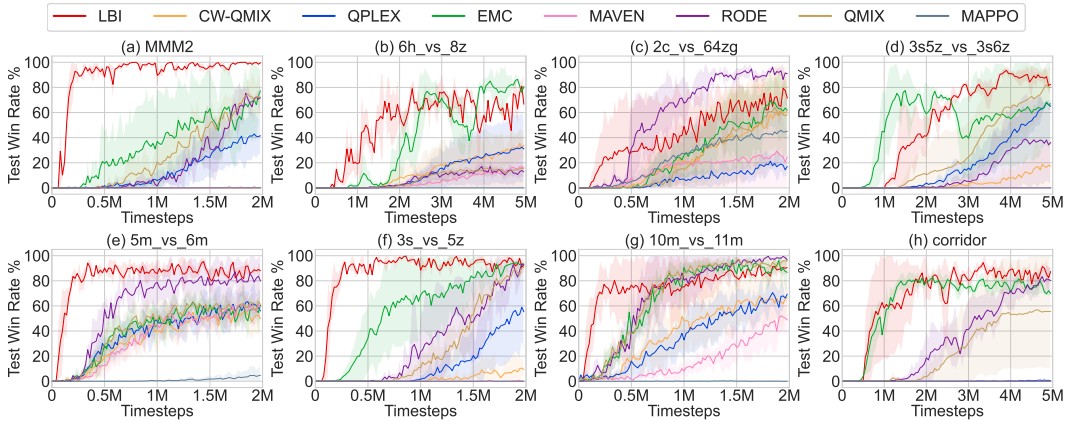

Figure 7: Performance comparisons between online learning methods using ground-truth rewards on the SMAC benchmark and LBI using the learned reward functions on the imagined world model.

In addition, we also show the qualitative comparison between the target and the generated sequences in Figure 8. Both trajectories are collected by running the same policy. We can see that the generated sequence can resemble the target one in most frames, but some differences exist in positions and health bars. However, compounding errors in the single-step model, which lead to physically implausible predictions, are not observed in the dynamics model generated by the causal transformer. For example, at the timestep of 10 in the MMM2 scenario, the generated frame does not contain the ally's Medivac, but we can see it in the following frames.

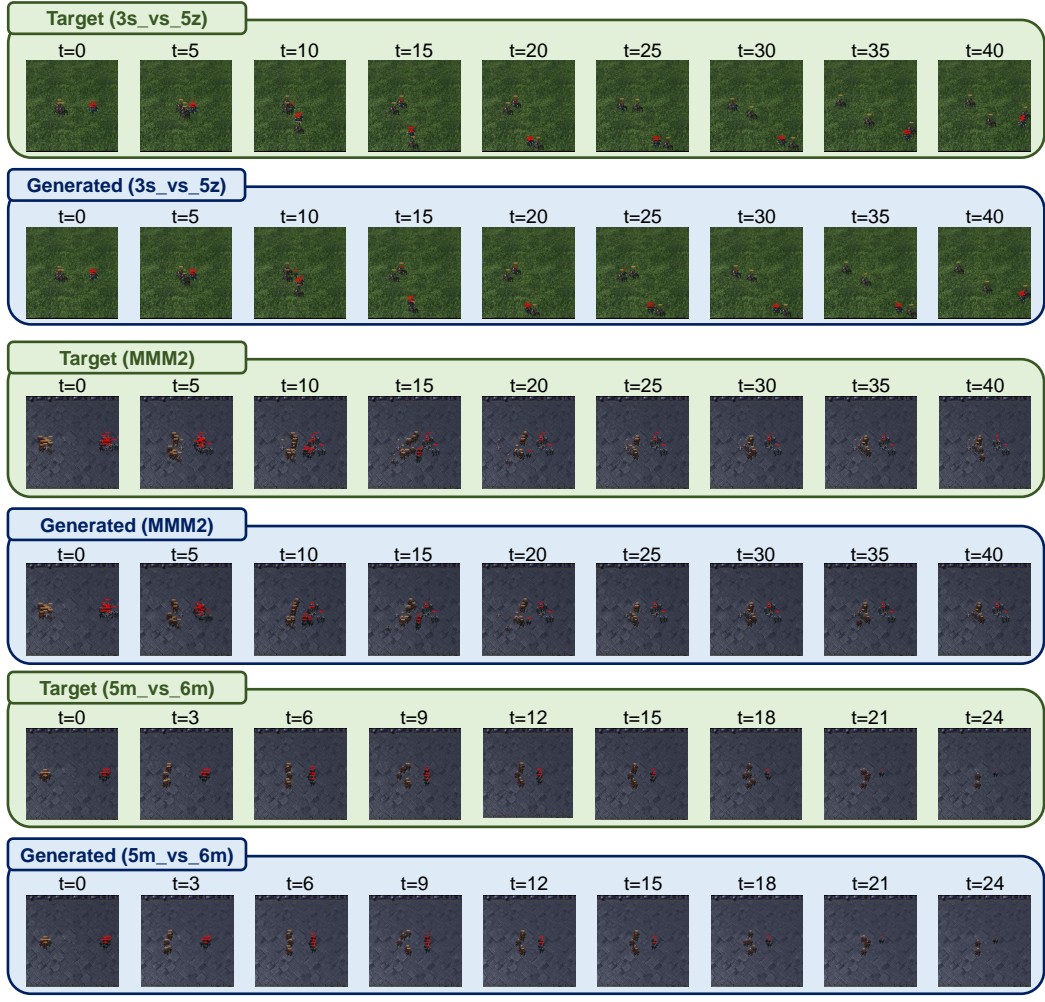

Figure 8: Comparisons of the target and the generated sequences across three different maps.

# F  Additional Related Work

**Transformer Model**   Several works have explored the integration of transformer models into reinforcement learning (RL) settings. We classify them into two major categories depending on the usage pattern. The first category focuses on representing components in RL algorithms, such as policies and value functions (Parisotto et al., 2020; Parisotto & Salakhutdinov, 2021). These methods rely on standard RL algorithms to update policy, where the transformer only provides a large representation capacity and improves feature extraction. Conversely, the second category aims to replace the RL pipeline with sequence modeling. They autoregressively generate states, actions, and rewards by conditioning on the desired return-to-go during inference (Chen et al., 2021; Lee et al., 2022; Reed et al., 2022). Due to its simplicity and potential generalization ability, this category is widely used in various domains, such as robotics control (Brohan et al., 2023a; Padalkar et al., 2023; Driess et al., 2023) and multi-agent reinforcement learning (Meng et al., 2023; Liu et al., 2024).

**Multi-agent Reinforcement Learning**   This section briefly introduces recent related work on cooperative multi-agent reinforcement learning (MARL). In the paradigm of centralized training with decentralized execution (CTDE), agents' policies are trained with access to global information in a centralized way and executed only based on local histories in a decentralized way (Oliehoek et al., 2008; Kraemer & Banerjee, 2016). One of the most significant challenges in CTDE is to ensure the correspondence between the individual $Q$-value functions and the joint $Q$-value function $Q_{tot}$, i.e., the Individual-Global Max (IGM) principle (Son et al., 2019). VDN (Sunehag et al., 2018)

and QMIX (Rashid et al., 2018) learn the joint $Q$-values and factorize them into individual $Q$-value functions in an additive and a monotonic fashion, respectively. Several works (Yang et al., 2020b,a; Wang et al., 2020b,c) have been proposed to improve the performance of QMIX, but as many previous studies pointed out, monotonic value function factorization limits the representational capacity of $Q_{tot}$ and fails to learn the optimal policy when the target $Q$-value functions are non-monotonic (Mahajan et al., 2019; Son et al., 2019; Rashid et al., 2020). To solve this problem, some recent works (Wang et al., 2020a; Mahajan et al., 2021) try to achieve the full representational capacity of $Q_{tot}$, while others prioritize the potential optimal joint action and learn a biased $Q_{tot}$.

Some independent learning algorithms have also proven robust in solving multi-agent cooperative tasks. Distributed $Q$-learning (Lauer, 2000) and Hysteretic $Q$-learning (Matignon et al., 2007) place more importance on positive updates that increase a $Q$-value estimate, which is similar to the weighting function in WQMIX. However, Wei & Luke (2016) prove that these methods are vulnerable towards misleading stochasticity and propose LMRL2, where agents forgive the other's miscoordination in the initial exploration phase but become less lenient when the visitation of state-action pair increases. MAPPO (Yu et al., 2022) applies PPO (Schulman et al., 2017) into MARL and shows strong empirical performance. However, Kuba et al. (2021) points out MAPPO suffers from instability arising from the non-stationarity induced by simultaneously learning and exploring agents. Therefore, they introduce the sequential policy update scheme to achieve monotonic improvement on the joint policy.

Learning communication protocols to solve cooperative tasks is one of the desired emergent behaviors of agent interactions. It has recently become an active area in MARL, such as learning to share observations (Das et al., 2019; Wang et al., 2019; Liu et al., 2020) and intentions (Kim et al., 2020; Böhmer et al., 2020; Wen et al., 2022; Liu et al., 2023).

