# OpenReview forum: "Grounded Answers for Multi-agent Decision-making Problem through Generative World Model"
_NeurIPS.cc/2024/Conference — NeurIPS 2024 poster_

### Official Review · Reviewer_yaXk · 2024-07-08

**Soundness:** 3
**Presentation:** 3
**Contribution:** 3
**Rating:** 7
**Confidence:** 3

**Summary:**

The paper proposes a new approach to multi-agent decision-making problems by integrating a language-guided simulator into the reinforcement learning pipeline.

Learning before Interaction (LBI) uses a world model to generate trial-and-error experiences and improve the answers to complex decision-making problems.

The world model consists of a dynamics model and a reward model, which are learned from expert demonstrations and task descriptions.

The LBI framework is evaluated on the StarCraft Multi-Agent Challenge benchmark and shows superior performance on both training and unseen tasks, generating consistent and explainable interaction sequences and rewards.

**Strengths:**

The proposed VisionSMAC dataset is a significant contribution, as it provides a large-scale, diverse, and realistic dataset for multi-agent decision-making problems in the StarCraft II environment.
The dataset is constructed by collecting offline trajectories using multi-agent exploration methods and then converting the state into images and languages through a parser, making it easy to create new content.

LBI is a novel approach that integrates a language-guided simulator into the reinforcement learning pipeline, enabling the generation of trial-and-error experiences and improving the answers to complex decision-making problems.

 the LBI framework and the VisionSMAC dataset provide a powerful tool for learning and reasoning in complex multi-agent decision-making problems.

**Weaknesses:**

The motivation for mentioning and comparing the answer of GPT-4, at the beginning, seems misleading. I thought that the proposed LBI would benefit the general response of LLMs on complex decision-making. However, it seems that the proposed method heavily relied on the gaming environment, and and result is limited to SMAC benchmark.

The dataset collection is a crucial part of developing a successful LBI, especially for the reward prediction part. However, it must be constructed via a world model (1) where the dynamic is explicit (2) simulation speed is fast enough to perform learning and interaction. How the generalizability of the proposed method?

**Questions:**

See weaknesses

**Limitations:**

no negative societal impact

---

> ### Author Rebuttal · Authors · 2024-08-07
>
> **1. Motivation seems misleading, and results are limited to the SMAC benchmark.**
>
> We aim to improve the output of large language models by constructing a differentiable interactive environment and then learning the policy on this interactive environment. The paper focuses on constructing such an environment, including the dynamics and reward models, and demonstrates the potential application of this framework.
>
> Due to the substantial computational resources required to train MARL methods, especially in most cooperative MARL environments that lack image inputs and have extremely low sampling efficiency, we have conducted experiments only on SMAC, which we have mentioned in the limitations section.
>
> Although SMAC cannot represent all possible multi-agent tasks, as a highly complex multi-agent reinforcement learning environment, SMAC involves heterogeneous agents, varying numbers and tactics, and different maps, making the learning process inherently challenging. This complexity can reflect the proposed algorithm's potential applications.
>
> The trick of masking out dead agents aims to simplify the inference difficulty of algorithms, and it seems task-specific. However, it could be a general trick for multi-agent environments with attrition, and it can also be applied to scenarios like MOBA or RTS games. In addition, developing such tricks might be a promising direction in generative models. For example, the outputs of large models are not always reliable, such as generating movements that do not satisfy physical constraints. Introducing human priors into scene prediction results could be an essential way to solve these issues. Methods like PINNs and Lagrangian conservation can impose constraints on certain predictions, thereby enhancing the reliability of the generated results. The tricks specific to the SMAC game in this paper can also be categorized as one of these methods.
>
> **2.  The generalizability of the proposed method.**
>
> The proposed method has a certain generalization ability in multi-agent environments.
>
> * Regarding the *dynamic is explicit* issue, if we understand correctly, you mean that the input for the reward is the entire trajectory rather than (s, a), as in most previous methods. This is necessary because we aim to achieve temporal utility distribution across the entire trajectory, which cannot be done by relying solely on (s, a), thus failing to address issues like delayed reward. During inference, we first run the dynamics model to obtain the entire reward-free trajectory and then run the reward model to label the rewards so the entire trajectory is known.
> If you mean that dynamics might not be deterministic, such as taking the same action in a given state leading to different following states and rewards, this is not considered in this paper. This is because most multi-agent reinforcement learning environments, including SMAC, Google Research Football, and Multi-agent Particle World, are deterministic. To address this, we may modify the loss function by introducing heteroscedastic aleatoric uncertainty estimation - we can model the predicted state and reward as distributions and estimate the mean and variance of both state and reward.
>
> * Concerning the issue of *simulation speed being fast enough for learning and interaction*, in our world model, dynamics is an auto-regressive generation process corresponding to interacting with the real environment (e.g., the function of env.step()). This is time-consuming and a common issue in reinforcement learning algorithms. The reward is predicted based on a bidirectional Transformer by inputting the entire trajectory, which is less time-consuming than the auto-regressive dynamics model. Additionally, since the world model is implemented entirely through neural networks, distributed data collection is easier than with the original game engine and does not require communication with the game engine (which is the main reason for the slow interaction in the SMAC environment). When transferred to a real environment, distributed data collection without interacting with the real environment offers higher speed and safety than online learning.

---

### Official Review · Reviewer_2fu7 · 2024-07-12

**Soundness:** 4
**Presentation:** 3
**Contribution:** 3
**Rating:** 5
**Confidence:** 4

**Summary:**

This paper proposes a method of improving the sample efficiency and performance of multi-agent reinforcement learning (RL) on the StarCraft Multi-Agent Challenge (SMAC) benchmark by learning to simulate the environment. The simulator consists of an image tokenizer, a dynamics model, and a reward function, all trained on data collected from interactions with the SMAC environment. A multi-agent policy then interacts with the simulator to obtain trajectories. These trajectories are then used to optimise the policy, but with a behaviour regularisation term towards the simulated trajectories. Evaluations on the SMAC benchmark show improved win rates and returns over existing online and offline multi-agent RL baselines. The paper also ablates the various components of the proposed method and provides visualisation of the simulated trajectories and their close match with actual trajectories.

**Strengths:**

Proposes a method of training multi-agent RL using a learned simulator. It carefully considers issues such as data collection (by creating a method of generating images from stored game state), image tokenization, using inverse RL to learn the reward function, and regularising the policy update. As a result they achieve quite impressive performance gains in SMAC when compared with prior work.

The paper thoroughly ablates the different components of their method and shows that while somewhat complex, each component is necessary for the best performance.

**Weaknesses:**

The paper only studies SMAC and no other multi-agent domain. It is hard to evaluate the generality of the method without other evaluations on standard benchmarks. Adding evaluation results from one or two more domains would help strengthen the generality of this method.

The write-up is a bit difficult to follow and contains a number of confusing parts (see also questions). The main figure, Figure 3, is never discussed in the main text. The part of Figure 1 on GPT-4 is not necessary for the paper.

A significant amount of detail is located in the Appendix, for example Appendix B on how the training data for the simulation components are collected, and Appendix E on the experimental details. Consider moving these sections into the main text.

The paper does not explain the methods used to collect the training data (EMC and IIE). To be better self contained, please give short explanations of these methods in the main text. Are they particularly tuned to the SMAC environment? What would have to change in domains where an effective exploration strategy is unknown?

**Questions:**

In the input notation of the dynamics model why is L_t time-varying across a trajectory? From the text L is constant across an entire trajectory, could the input contain more than 1 trajectory? The notation could be improved.

Could you explain in more detail how hindsight-relabelling is done and why it is necessary for reward function training? Is it the same or different from the text label used on the same set of collected trajectories as in dynamics model training? Is it only for data collected from the simulator?

Clarifications for equation (1): What is D? Why does the f term have \tau as an input and an index i? Where do they come from? Is there a missing sum(s)? In the description it says that term is “over all possible actions”, what does this mean? Its inputs are trajectories and aren’t guaranteed to contain all actions?

Why did the authors choose to only focus on a single multi-agent domain and not evaluate the method on any other, even toy, multi-agent benchmarks?

Why do some tables report win rate and some rewards?

Figure 8, does LBI timesteps account for the number of time steps in the data collected to train the simulator components? Would it be more accurate to shift the LBI curve to the right by that amount?

**Limitations:**

Limitations have been adequately addressed.

---

> ### Author Rebuttal · Authors · 2024-08-07
>
> **1. The exploration methods (EMC and IIE) for offline data collection**
>
> EMC is curiosity-based to boost multi-agent exploration for novel states, i.e., reduce out-of-distribution regions in the offline datasets. IIE belongs to go-explore algorithms that perform imagination without exploration, initialize agents at the end state of the imagination, and then start to explore. This method can reduce the multi-agent exploration space in the time dimension. We chose them to collect data because they have tuned in to the SMAC benchmark and have performed well in super hard maps.
>
> When collecting offline datasets in other MARL tasks, we must choose appropriate exploration methods based on the task properties. If the task involves many decision steps (long-horizon task), go-explore methods can be more effective; if the action space of each agent is huge, we can perform role-based learning (RODE) to decompose the action space; if we want to cover the entire state space, we can use curiosity-based exploration methods (EMC) or hierarchical exploration (MAVEN). Since the joint action and state space are huge in SMAC, we chose EMC to reduce the impact from OOD regions and used IIE to reduce exploration space.
>
> **2. $L_t$ in the dynamics model**
>
> Thanks for pointing it out. The input only contains one target sequence. In general, $L_t$ should reflect the task and the environment, which can be time-varying. The language description for the trajectory from $s_t$ and the terminated (goal) state $s_T$, e.g., return-to-go). However, in practice, we simplify this language description to be time-invariant, which is always the language description of terminated (goal) state $s_T$. We have corrected this for better understanding.
>
> **3. Hindsight-relabelling in the reward model**
>
> We use the same text label for the reward and dynamics models to specify the trajectory and the reward generation. We use hindsight learning to provide a specific condition for reward generation to avoid the influence of suboptimal data in the offline dataset. Without this condition, the reward model would learn to approximate all possible trajectories in the dataset. Due to the presence of suboptimal trajectories, this would negatively impact the reward model's learning. We train the reward model using conditional generation to learn the reward functions corresponding to different outcomes. During inference, we set a condition, such as asking the agents to completely eliminate the enemy, to extract the reward function corresponding to a successful trajectory from the reward model.
>
> In policy models, this type of condition is usually set as return-to-go (RTG) to achieve desired effects. In contrast, we use language descriptions to set the conditions, which allows for the generation of an interactive simulator and enables the model to have multimodal processing capabilities.
>
> **4. Only focus on the SMAC benchmark**
>
> Due to the substantial computational resources required to train MARL methods, especially in most cooperative MARL environments that lack image inputs and have extremely low sampling efficiency, we have conducted experiments only on SMAC, which we have mentioned in the limitations section.
>
> Although SMAC cannot represent all possible multi-agent tasks, as a highly complex multi-agent reinforcement learning environment, SMAC involves heterogeneous agents, varying numbers and tactics, and different maps, making the learning process inherently challenging. This complexity can reflect the proposed algorithm's potential applications.
>
> **5. Why do some tables report win rates and some rewards?**
>
> In SMAC, a slight difference in return (19 vs. 20) can result in a significant difference in win rate (0% vs. 90%). If we only show the win rate, the results for most current offline methods can always be 0, making comparisons less meaningful. Using cumulative rewards as a metric is also reasonable because reinforcement learning inherently optimizes cumulative rewards.
>
> **6. Consider the number of time steps in the data collection.**
>
> First, the pre-trained world model is an interactive simulator, and we compare it with the real environment, not the online methods themselves. This experiment aims to demonstrate that training a randomly initialized joint policy on the *learned world model* is faster than training it in the *real environment* - the learned world model (interactive simulator), specifically the learned transition and reward functions, is accurate. So, we do not shift the LBI curve to the right by the number of time steps in the data collection.
>
> **7. Clarifications for equation (1).**
>
> $D$: the demonstrations provided by an expert policy.
>
> The index $i$ of $\tau$ in the $f$ term: Thanks for pointing this out. We perform this regularization on the predicted rewards from the expert demonstrations and the data collected by the MA-SAC policy. Namely, in the $f$ term, $\tau\sim \\{\boldsymbol{\pi}^\theta,D\\}$, $i$ and $t$ should be added in the agent and the timestep dimensions.
>
> Over all possible actions: This is an implenmentation trick. In practice, the reward model $RM$ takes the action-free trajectory and outputs the rewards for all actions of each agent. Namely, the individual reward head outputs $|U|$ rewards for each agent, where $|U|$ is the action space. Then, we choose the reward corresponding to the action using the torch.gather function. We add the L2 penalization for these $|U|$-dim predicted rewards to avoid overestimation on out-of-distribution actions. For example, if the offline dataset does not contain the action $u_t^i$, then the predicted reward of $u_t^i$ does not have a target, introducing potential risks for policy learning.

---

> > ### Comment · Reviewer_2fu7 · 2024-08-13
> > **Reply**
> >
> > I thank the authors for the detailed rebuttal. Most of my concerns has been addressed. However, one key concern of the limiting scope of the work only on the SMAC benchmark (which it seems is also shared with other reviewers) has not been addressed. I will maintain my original rating.

---

### Official Review · Reviewer_sAvh · 2024-07-13

**Soundness:** 3
**Presentation:** 2
**Contribution:** 3
**Rating:** 5
**Confidence:** 4

**Summary:**

This paper introduces a pipeline called Learning before Interaction, a approach to improve generative models' answers for multi-agent decision-making problems. The key idea is to integrate a language-guided simulator into the multi-agent reinforcement learning pipeline.

The main contributions are
- Creating VisionSMAC, a dataset that pairs states with images and task descriptions for the StarCraft Multi-Agent Challenge
- Training an interactive simulator consisting of a dynamics model and a reward model

The pipeline use the simulator to train a multi-agent policy through simulated interactions and generate answers by running the trained policy on the simulator.

**Strengths:**

- LBI outperforms existing offline and imitation learning methods on unseen tasks in the SMAC. (good zero-shot generalization ability)
- The approach produces explainable reward functions at critical states, which can help in understanding the decision-making process

**Weaknesses:**

- (Not mentioned in the paper) Training an interactive simulator with both dynamics and reward models likely requires significant computational resources
- It would be great if there are environments other than SMAC tested. Otherwise, the scope might be too narrow.
- The approach requires a substantial amount of offline data to train the world model and compared with online MARL methods which is not fair comparison. The offline data is also not always available.
- The ablation study shows that using image reference doesn't significantly improve performance, suggesting that this aspect of the model might not be fully utilized or optimized.

**Questions:**

- What is the world model architecture and how do you train it?
- Could you please write an algorithmic block for both inference and training? I found there are still many processes unclear after reading the paper. It will be very helpful if you can upload a sample code (please be careful about the anonymity in the code).

**Limitations:**

The authors do not mention a limitation.

---

> ### Author Rebuttal · Authors · 2024-08-07
>
> **1. It requires significant computational resources**
>
> We listed the hyperparameters in the interactive simulator in Tables 8 and 9, which requires computational resources. However, we believe this framework is meaningful. We utilize the generative model as an interactive environment for RL to enhance the quality of large models' responses to decision-making problems. RL requires significant interaction with the environment. Since the world model is implemented entirely through neural networks, distributed data collection is easier than with the original game engine and does not require communication with the game engine (which is the main reason for the slow interaction in the SMAC environment). When transferred to a real-world task, distributed data collection without interacting with the real-world environment offers higher speed and safety than online learning.
>
> **2. The scope might be too narrow.**
>
> Due to the substantial computational resources required to train MARL methods, especially in most cooperative MARL environments that lack image inputs and have extremely low sampling efficiency, we have conducted experiments only on SMAC, which we have mentioned in the limitations section.
>
> Although SMAC cannot represent all possible multi-agent tasks, as a highly complex multi-agent reinforcement learning environment, SMAC involves heterogeneous agents, varying numbers and tactics, and different maps, making the learning process inherently challenging. This complexity can reflect the proposed algorithm's potential applications.
>
> **3. Unfair comparisons.**
>
> First, the pre-trained world model is an interactive simulator, and we compare it with the real environment, not the online methods themselves. This experiment aims to demonstrate that training a joint policy on the learned world model is faster than training it in the real environment - the learned world model (interactive simulator), specifically the learned transition and reward functions, is accurate.
>
> Second, this paper mainly compares offline Q-learning and inverse learning methods, demonstrating their generalization ability to unseen tasks.
>
> **4. The effectiveness of image reference.**
>
> Using images as a reference can improve performance (LBI and LBI-wo-IR), although less significant than the residual term (LBI and LBI-wo-RT). However, we demonstrated that the error and return when using real images (LBI-GTI) are similar to LBI, indicating that the benefit of image reference is limited rather than suggesting that the model is not optimized. The visual results show that the generated images are similar to the real ones.
>
> It is important to note that the model uses images and text as initial inputs and then performs autoregressive prediction, making images necessary for this task.
>
> **5. The world model architecture.**
>
> The world model consists of a VQ-VAE model for the image tokenizer, a casual transformer model for dynamics prediction, and a bi-directional transformer model for reward prediction. We build the VQ-VAE and the transformer model implementation based on (https://github.com/Project-MONAI/GenerativeModels/blob/main/generative/networks/nets/vqvae.py) and Decision Transformer (which borrows the code from minGPT https://github.com/karpathy/minGPT), the publicly available re-implementation of VQ-VAE and GPT2, respectively. The dynamics and reward models share the same transformer architecture, where the reward model does not use causal masks. We mentioned it in Appendix D.3.
>
> **6. Traning and inference.**
>
> The model is trained in two phases following a standard autoregressive video generation pipeline: We train the image tokenizer first, which is used for the dynamics and reward models. We then train the dynamics model and reward model separately.
>
> *VQ-VAE*: (1) sample a batch of images; (2) calculate the reconstruction and the quantization error; (3) optimize the VQ-VAE model.
>
> *Dynamics model*: (1) sample a batch of trajectories from the offline buffer; (2) calculate the output for each token; (3) optimize the dynamics model by minimizing the cross-entropy loss for actions and mean-squared error for others.
>
> *Reward model*:
>
> * Randomly initialize a MA-SAC policy network
>
> * While True do:
>
>     + Use MA-SAC to collect a batch of trajectories $B$ from the ground-truth environment
>
>     + Sample a batch of trajectories $D$ from the offline buffer
>
>     + Mask out rewards of the trajectories $B$ and $D$
>
>     + Generate the reward via the reward model
>
>     + Update the reward model by maximizing the predicted reward for $D$, and minimizing the predicted rewards for $B$;
>
>     + Update MA-SAC policy network for k-step (k=5).
>
> *Inference*:
>
> * Randomly initialize a policy network
>
>     + While True do:
>
>         * Given an initialized image and a task description, calculate the image embedding via the VQ-VAE model and the language embedding via a pre-trained language tokenizer named bert-base-uncased
>
>         * While all agents or all enemies are alive do
>
>             + Generate state via dynamics model
>
>             + Generate joint observations via dynamics model
>
>             + Generate actions (only for the behavior-regularization) via the dynamics model
>
>             + Obtain joint actions (for dynamics prediction) via current policy network (using $\epsilon$-greedy)
>
>             + Generate the next image embedding
>
>             + Concatenate the language embedding
>
>         * Add masks into the reward-free trajectory for the reward prediction
>
>         * The reward model takes the entire trajectory as input and generates corresponding rewards
>
>         * Store the whole trajectory with actions for the behavior regularization in replay buffer
>
>         * If the trajectories in replay buffer > batch size then
>
>             + Sample a random batch of trajectories
>
>             + Calculate the TD loss with the behavior regularization for the policy network
>
>             + Update policy network

---

> > ### Comment · Reviewer_sAvh · 2024-08-11
> >
> > Thanks for the comments. I am happy to raise the score.

---

### Official Review · Reviewer_Frvn · 2024-07-15

**Soundness:** 2
**Presentation:** 2
**Contribution:** 2
**Rating:** 6
**Confidence:** 4

**Summary:**

The paper presents a novel approach called Learning before Interaction (LBI) that integrates a language-guided simulator into the multi-agent reinforcement learning process to enhance the quality of solutions for complex decision-making problems. The LBI paradigm uses a world model comprising a dynamics model and a reward model to simulate interactions and improve policy learning. The authors introduce new offline MARL datasets for the StarCraft Multi-Agent Challenge and demonstrate LBI's effectiveness through notable performance on training and unseen tasks.

**Strengths:**

1. Endeavors in building a feasible task-guided multi-agent world model. By introducing task description containing thorough information and the image of states of the task, this work empirically manage to build a world model being adaptive to various tasks in the training data distribution.

2. Designs for separate reward model and dynamics model. By decoupling the prediction of future observations and the prediction of rewards, this work can learn an efficient reward function involving reasonable credit assignments by adopting the inverse RL.

**Weaknesses:**

1. Trivial framework designed for multi-agent world model. Overall, the method proposed in this work seems not to capture the distinct characteristics in the context of multi-agent scenarios as compared to single-agent scenarios. The built world model appears to be based on reduce multi-agent problems into a single-agent problem with joint observations and joint actions. What are the benefits of this approach? Why is it necessary to adopt this simplification? The article does not provide specific evidence or thorough analysis to support these architecture design decisions.

2. Lack of the investigation and citation of related multi-agent world model literature. Given that the method proposed in this work derives the optimal policy via learning in the imaginations of the world model, it is essential to elucidate the differences between this approach and those presented in [1, 2, 3, 4] and to conduct performance comparisons. Notably, the architecture closely resembles that of [1] in the absence of reward prediction. However, the entire paper lacks citations and clarifications regarding these similarities.

3. Unfair comparisons. According to the contents in the appendix, the average returns are almost 20 in all offline dataset of ten maps, which are nearly optimal in SMAC environments since the maximum return of an episode is 20 in almost all SMAC maps. Thus with a pretrained world model which has full access to the optimal demonstrations, the comparison with online methods is definitely not meaningful and convincing at all.

4. Heavily handcrafted processing. “Since it is possible for specific agents to become inactive before the game terminates, we mark the terminated timestep for each agent and enemy once its predicted health is less than zero …” To identify the predicted health, the algorithm has implicitly already known the specific meaning of the value in some certain dimensions in the observation, which is obviously induced from the prior knowledge or predefined rules from human beings. Unfortunately, such an operation makes this work only narrowed in the domain of SMAC.

[1] Micheli, Vincent, Eloi Alonso, and François Fleuret. "Transformers are Sample-Efficient World Models." The Eleventh International Conference on Learning Representations.

[2] Egorov, Vladimir, and Alexei Shpilman. "Scalable Multi-Agent Model-Based Reinforcement Learning." Proceedings of the 21st International Conference on Autonomous Agents and Multiagent Systems. 2022.

[3] Zhang, Yang, et al. "Decentralized Transformers with Centralized Aggregation are Sample-Efficient Multi-Agent World Models." arXiv preprint arXiv:2406.15836 (2024).

[4] Liu, Qihan, et al. "Efficient Multi-agent Reinforcement Learning by Planning." The Twelfth International Conference on Learning Representations.

**Questions:**

See weaknesses.

---

> ### Author Rebuttal · Authors · 2024-08-07
>
> **1&2. Trivial framework and lack of investigation of related multi-agent world model literature.**
>
> If I understand correctly, the distinct characteristic of multi-agent scenarios you mentioned should be decentralized execution, which is a common way to simplify computational demands during the execution and thus enhance the scalability in practical applications. However, centralized training is generally required during the training process because agents learn the joint policy in a non-stationary environment. For example, QMIX [1] uses monotonic value decomposition to learn the joint Q-value function, and HAPPO [2] proposes sequential learning and multi-agent advantage decomposition. Otherwise, using individual Q-learning could lead the algorithm into local optima (unless we use some independent learning methods like lenient learning or design individual reward functions).
>
> The most significant difference is that our aim is to pre-train a world model and utilize it to train the policy model rather than directly using it as a part of the policy model.
>
> * [3] is an image-based single-agent world model, while ours is a multi-modal multi-agent world model. This dynamics model structure is general and is also used in GAIA, DT, TT, and Genie, which we mentioned in the related work on model-based and Transformer methods.
>
> * [4][5][6] are online learning methods that use the world model to generate imagined future trajectories at a given state for planning or better decision-making. Our goal is to learn an interactive simulator from offline datasets.
>
> * [3][4][5][6] suppose the reward is given, and [4][5][6] enable communication during execution to add necessary inputs for the agents. We suppose the reward is unknown and use inverse RL to learn the reward, and one can make use of different MARL methods (e.g., model-based or model-free, value-based or policy-based) to learn the policy on the proposed world model.
>
> For the world model, since we do not use it in the policy execution but treat it as a simulator – the learned policy can still be decentralized. To provide enough information for the input, we train directly using joint behaviors and observations.
>
> Thanks for pointing out these works. We would like to add them to the World Model and Multi-agent Reinforcement Learning subsection.
>
> **3. Unfair comparisons.**
>
> First, the pre-trained world model is an interactive simulator, and we compare it with the real environment, not the online methods themselves. This experiment aims to demonstrate that training a joint policy on the learned world model is faster than training it in the real environment - the learned world model (interactive simulator), specifically the learned transition and reward functions, is accurate. Designing a multi-agent environment, precisely the reward function, is complex and requires much prior. Our framework shows that utilizing the world model (a causal transformer for dynamics and a bidirectional transformer for reward) can successfully perform inverse reinforcement learning and learn meaningful reward functions.
>
> Second, this paper mainly compares offline Q-learning and inverse learning methods, demonstrating their generalization ability to unseen tasks. The comparison with online MARL is only briefly mentioned in the main text, with detailed results provided in the appendix.
>
> **4. Heavily handcrafted processing - the agent death masks.**
>
> This design aims to simplify the inference difficulty of algorithms, and it seems task-specific. However, it could be a general trick for multi-agent environments with attrition, and it can also be applied to scenarios like MOBA or RTS games. For example, MAPPO [7] also uses such death masking for better value estimation.
>
> In addition, developing such tricks might be a promising direction in generative models. For example, the outputs of large models are not always reliable, such as generating movements that do not satisfy physical constraints. Introducing human priors into scene prediction results could be an essential way to solve these issues. Methods like PINNs and Lagrangian conservation can impose constraints on certain predictions, thereby enhancing the reliability of the generated results. The tricks specific to the SMAC game in this paper can also be categorized as one of these methods.
>
> [1] Tabish Rashid, et al. Qmix: Monotonic value function factorisation for deep multi-agent reinforcement learning.
>
> [2] Jakub Grudzien Kuba, et al. Trust region policy optimisation in multi-agent reinforcement learning.
>
> [3] Micheli, Vincent, Eloi Alonso, and François Fleuret. Transformers are Sample-Efficient World Models.
>
> [4] Egorov, Vladimir, and Alexei Shpilman. Scalable Multi-Agent Model-Based Reinforcement Learning.
>
> [5] Zhang, Yang, et al. Decentralized Transformers with Centralized Aggregation are Sample-Efficient Multi-Agent World Models.
>
> [6] Liu, Qihan, et al. Efficient Multi-agent Reinforcement Learning by Planning.
>
> [7] Chao Yu, et al. The surprising effectiveness of ppo in cooperative multi-agent games.

---

> > ### Comment · Reviewer_Frvn · 2024-08-10
> > **Response**
> >
> > 1. First, different from the single-agent scenario, the scalability issue stemming from increasing agents and the non-stationarity issue arising from the continuously changing environment of individual agents, are inherent in the multi-agent dynamics. However, the authors gave little explanation on whether and how this work can tackle these challenges.
> > 2. The author may not carefully investigate the related works [1-2]. MAMBA and MARIE adopted a learning policy in the imagination scheme, enabling them to seamlessly incorporate different MARL methods. Their induced policies also remained decentralized.
> > 3. I don't think the current experiment is aligned to demonstrate that training a policy in the world model is faster than training it in a real environment. In contrast, like how IRIS [3] delivered this similar insight via its experiments, it would be better to carry out experiments with a low data regime like Atari-100K to demonstrate the aim.
> > 4. In terms of the handcrafted processing for obtaining agent death masks, it makes the work only applicable to the limited SMAC domain, weakening the contribution. Moreover, a prediction towards the agent death masks in the world model can better validate how accurate the world model, as the world model needs to understand the underlying game mechanics in SMAC. The handcrafted processing oversimplifies the learning.
> >
> > [1] Egorov, Vladimir, and Alexei Shpilman. Scalable Multi-Agent Model-Based Reinforcement Learning.
> >
> > [2] Zhana, Yang, et al. Decentralized Transformers with centralized Aggregation are Sample-Efficient Multi-Agent World Models
> >
> > [3] Micheli. Vincent, Eloi Alonso, and Francois Fleuret, Transformers are Sample-Efficient World Models

---

> > > ### Author Response · Authors · 2024-08-12
> > > **Rebuttal during the discussion**
> > >
> > > We appreciate the reviewer’s feedback!
> > >
> > > **1. The non-stationarity and the scalability issue.**
> > >
> > > We provide the state and joint actions as the input to train the world model (avoid the non-stationarity) and utilize the Transformer model (GPT2-like dynamics and Bert-like reward models) to provide enough representational capacity, where their attention mechanism performs feature selection and aggregation (for the scalability).
> > >
> > > To further elucidate our approach, we list the following paradigms:
> > >
> > > *(1) Decentralized training with decentralized execution*: While this paradigm excels in scalability during execution, it often leads to non-stationarity as agents treat others as part of the environment.
> > >
> > > *(2) Centralized training with decentralized execution*: It maintains execution scalability while introducing a centralized mechanism (e.g., a mixing network to aggregate the individual Q-values) to mitigate the non-stationarity problem.
> > >
> > > *(3) Communication-enabled training and execution*: Each agent learns and executes its policy based on their observations and (weighted) messages from others (e.g., actions, observations). This paradigm creates a trade-off between interaction complexity and representational capacity, a great compromise for non-stationarity and scalability issues.
> > >
> > > *(4) Centralized training with centralized execution*: While this paradigm eliminates the non-stationarity issue, it suffers from limited scalability during execution.
> > >
> > > Our world model serves as the environment, which leverages the Transformer model’s attention mechanism to capture interdependencies among agents and timesteps, similar to communication-enabled paradigms (e.g., MAT [1]). The concurrent work [2] also proposes a Perceiver Transformer as a communication module, which utilizes the querying and self-attention mechanisms to perform agent-wise aggregation, transforming the joint representation sequence (joint observations and actions) into lower-dimensional features. The difference is that our world model integrates these processes into the large transformer model.
> > >
> > > **2&3. MAMBA and MARIE, and the low data regime.**
> > >
> > > We acknowledge that MAMBA [3] and MARIE [2] are indeed Dyna-style algorithms that learn policy in the imagination, and their induced policies are decentralized. We would like to clarify our stance:
> > >
> > > * MAMBA and MARIE assume that the reward function is presented in the replay buffer. They train the reward predictor using log-likelihood and cross-entropy loss, respectively. **Conversely**, we suppose the reward is unknown in the offline dataset. We construct the reward function using a bidirectional Transformer, which maximizes the likelihood of trajectories in hindsight relabeled expert demonstrations.
> > >
> > > * MAMBA and MARIE are online model-based RL methods. They explore the environment using a policy, store the collected data in a replay buffer, update the world model with samples from this buffer, and use rollouts from the world model to refine the policy. They can benefit from interacting with the real environment, allowing them to update and rectify their world models continuously - e.g., if agents imagine trajectories in out-of-distribution regions that lead to suboptimal updates, they can correct this during exploration by gathering more data in these regions. Consequently, they conducted experiments in low-data regimes like Atari-100K to show their data efficiency.
> > >
> > >   **In contrast**, our dynamics model is trained in an offline manner via a static, offline dataset. Unlike online methods, no exploration is available for the model to rectify itself, leading to the distributional shift issue - where the deployment scenario may differ significantly from the historical dataset regarding state and action space. To mitigate this, we employ a behavior regularization term to encourage exploitation, specifically by discouraging agents from executing out-of-distribution behaviors during inference.
> > >
> > > **4. The agent death masks.**
> > >
> > > Thank you for your insightful suggestion. We would like to use a binary gate function for each agent to replace the handcrafted death masks and learn it via binary cross-entropy loss in a supervised manner. We acknowledge the potential for further generalization.
> > >
> > > However, we respectfully disagree with the notion that the current approach oversimplifies the learning process. As shown in Fig.4, the learned world model captures the dynamics and learns a meaningful reward function even in this complex scenario. This demonstrates that the model correctly understands the transitions within the game, validating its ability to grasp the underlying mechanics of SMAC.
> > >
> > > [1] Wen, Muning, et al. Multi-agent reinforcement learning is a sequence modeling problem.
> > >
> > > [2] Zhana, Yang, et al. Decentralized Transformers with centralized Aggregation are Sample-Efficient Multi-Agent World Models.
> > >
> > > [3] Micheli. Vincent, Eloi Alonso, and Francois Fleuret, Transformers are Sample-Efficient World Models.

---

### Decision · Program_Chairs · 2024-09-25

**Decision:**

Accept (poster)

**Comment:**

This paper introduces the LBI framework, which employs a world model consisting of dynamics and reward components to simulate interactions and enhance policy learning. A multi-agent policy interacts with this simulator to generate trajectories. These trajectories are subsequently utilized to optimize the policy.

**strengths**

* Novel idea of integrating a language-guided simulator into RL pipeline

* VisionSMAC: a large-scale, diverse, and realistic dataset for multi-agent decision-making problems in the StarCraft II environment

* Extensive experiments and analysis on different components

**weaknesses/suggestions**

* Discussions on computational cost

* Limited domain (only SMAC)

I think the paper makes a nice contribution that the community will find valuable. However, I encourage the authors to think carefully about how to reflect the comments or resolve the questions from reviewers in the camera ready version.